# Incidence and Survival Trends of Pancreatic Cancer in Girona: Impact of the Change in Patient Care in the Last 25 Years

**DOI:** 10.3390/ijerph17249538

**Published:** 2020-12-19

**Authors:** Adelaida García-Velasco, Lluís Zacarías-Pons, Helena Teixidor, Marc Valeros, Raquel Liñan, M. Carmen Carmona-Garcia, Montse Puigdemont, Walter Carbajal, Raquel Guardeño, Núria Malats, Eric Duell, Rafael Marcos-Gragera

**Affiliations:** 1Medical Oncology Department, Josep Trueta Universitary Hospital, Catalan Institute of Oncology, Av de França, 17007 Girona, Spain; rlinan@iconcologia.net (R.L.); ccarmona@iconcologia.net (M.C.C.-G.); wcarbajal@iconcologia.net (W.C.); rguardeno@iconcologia.net (R.G.); 2Descriptive Epidemiology, Genetics and Cancer Prevention Group, Biomedical Research Institute (IDIBGI), C/Dr. Castany, s/n, 17190 Salt, Spain; mpuigdemont@iconcologia.net (M.P.); rmarcos@iconcologia.net (R.M.-G.); 3Epidemiology Unit and Girona Cancer Registry, Oncology Coordination Plan, Department of Health, Autonomous Government of Catalonia, Catalan Institute of Oncology, Av. França, s/n, 17004 Girona, Spain; lzacarias@idibgi.org (L.Z.-P.); helenateixipuig@gmail.com (H.T.); marcvaleros97@gmail.com (M.V.); 4Genetic and Molecular Epidemiology Group, Spanish National Cancer Research (CNIO) and CIBERONC, 28029 Madrid, Spain; nmalats@cnio.es; 5Unit of Biomarkers and Susceptibility, Oncology Data Analytics Program, Catalan Institute of Oncology (ICO), Colorectal Cancer Group, ONCOBELL Program, Bellvitge Biomedical Research Institute (IDIBELL), L’Hospitalet de Llobregat, Consortium for Biomedical Research in Epidemiology and Public Health (CIBERESP), 08908 Barcelona, Spain; eduell@iconcologia.net; 6Consortium for Biomedical Research in Epidemiology and Public Health (CIBER Epidemiología y Salud Pública, CIBERESP), 28029 Madrid, Spain

**Keywords:** pancreatic cancer, incidence, relative survival, pancreatic neuroendocrine tumours

## Abstract

(1) Background: We investigated the incidence and survival trends for pancreatic cancer (PC) over the last 25 years in the Girona region, Catalonia, Spain; (2) Methods: Data were extracted from the population-based Girona Cancer Registry. Incident PC cases during 1994–2015 were classified using the International Classification of Diseases for Oncology Third Edition (ICD-O-3). Incidence rates age-adjusted to the European standard population (ASR_E_) and world standard population (ASR_W_) were obtained. Trends were assessed using the estimated annual percentage of change (EAPC) of the ASR_E13._ Observed and relative survivals (RS) were estimated with the Kaplan–Meier and Pohar Perme methods, respectively; (3) Results: We identified 1602 PC incident cases. According to histology, 44.4% of cases were exocrine PC, 4.1% neuroendocrine, and 51.1% malignant-non-specified. The crude incidence rate (CR) for PC was 11.43 cases-per-100,000 inhabitants/year. A significant increase of incidence with age and over the study period was observed. PC overall 5-year RS was 7.05% (95% confidence interval (CI) 5.63; 8.84). Longer overall survival was observed in patients with neuroendocrine tumours (5-year RS 61.45%; 95% CI 47.47; 79.55). Trends in 5-year RS for the whole cohort rose from 3.27% (95% CI 1.69–6.35) in 1994–1998 to 13.1% (95% CI 9.98; 17.2) in 2010–2015; (4) Conclusions: Incidence rates of PC in Girona have increased in the last two decades. There is a moderate but encouraging increase in survival thorough the study period. These results can be used as baseline for future research.

## 1. Introduction

Pancreatic cancer (PC) is a major health problem in developed countries due to its high mortality rate and its increasing incidence [1]. It used to be an uncommon neoplasm in the European Union (EU) but now it ranks as the 7th most common form of cancer, representing 3.0% of all new cancer cases [2]. Cancer incidence projections of GLOBOCAN (Global Cancer Observatory) 2018 point out that the number of new cases will rise in the whole EU (+32%) by 2040, and up to +46% in Spain [3]. It is already the fourth leading cause of cancer death in Western countries, the 3rd in Spain [4], and in the next 20 years its impact is expected to increase up to being the second leading cause of cancer death and the main cause of lost years of quality-adjusted life (QALYs) [5].

More than 90% of malignant tumors of the pancreas arise from the exocrine pancreas, with pancreatic ductal adenocarcinoma representing 85% of cases [6]. Its prognosis is ominous, with overall survival rates under 20% one year after diagnosis and <10% at 5 years [7,8,9,10]. This has been attributed, among other causes, to the premature vascular, lymphatic and perineural spread of these tumors, which makes that most of the patients present disseminated disease at diagnosis [11]. Improvements in diagnostic tools and systemic treatments have been achieved in the last two decades benefitting a small proportion of patients and, therefore, with only minimal impact on mortality. In this context, PC represents a major challenge for cancer research and has fostered growing interest by clinicians and epidemiologists.

PC etiology is still not well known. Established risk factors are smoking, high alcohol consumption, diabetes mellitus, obesity, chronic pancreatitis, mucinous intra-ductal neoplasia (IPMN), familial aggregation of PC and some hereditary syndromes (i.e., familial multifocal atypical melanoma, Peutz–Jeghers syndrome and hereditary breast and ovarian cancer associated with BRCA mutations) [6]. Furthermore, up to 40 low-penetrance genetic variants have been associated with sporadic PC risk based on GWAS (Genome-wide association study) in consortia of multiple PC studies. However, these factors do not fully explain the variation in risk for PC, and familial risk and genetic syndromes account for less than 10% of PC.

We conducted a population-based study of patients diagnosed with PC in Girona, a north-eastern province of Spain with one of the oldest population cancer registries in the country with regular data quality review, to analyze the incidence and survival trends for PC in the region.

## 2. Materials and Methods

### 2.1. Data Source

Data were extracted from the population-based Girona Cancer Registry (GCR)—in the northeast of Spain (covering a population of 739,607 inhabitants in 2015)—during 1994–2015. Cases were registered according to the International Agency for Research on Cancer (IARC) guidelines with a completeness of 96.3%.

### 2.2. Study Population

The study included cases with PC (International Classification of Diseases for Oncology Third Edition (ICD-O-3) histological codes: 8000, 8010, 8020, 8041, 8070, 8140, 8150, 8151, 8153, 8240, 8246, 8249, 8260, 8452, 8453, 8470, 8480, 8481, 8490, 8500, 8510, 8550, 8560 and 8971; topography C25) [12]. We excluded patients with tumors in situ and histologically verified stromal tumors, lymphomas and sarcomas (*n* = 3 cases). Also, patients aged less than 15 years old at the date of PC diagnosis were excluded. All cases were retrospectively revised individually. Tumors were categorized by histological subtypes following the 4th edition of World Health Organisation (WHO) classification of tumors of the digestive system [13], to ensure comparability. For the analysis, two main histological subgroups were considered: pancreatic neuroendocrine tumours (PNET) and non-neuroendocrine PC (non-PNET), besides, a third subgroup (non-HC PC) includes all cases without microscopic verification (Appendix A). The study population was further stratified by sex and grouped by age at diagnosis into five categories: (15–44 y), (45–54 y), (55–64 y) and elderly persons (65–74 y and ≥75 y).

The primary site was defined following the ICD-O-3 codes C25.0–C25.9: Head of pancreas, (C25.0), body of pancreas (C25.1), tail of pancreas (C25.2), pancreatic duct (C25.3), “others” including islets of Langerhans (C25.4), other specified parts of pancreas (C25.7), overlapping lesion of pancreas (C25.8) and pancreas, NOS (C25.9). Due to the low numbers of cases we jointly analysed the following codes: C25.3 (ducts), C25.4 (islets+), C25.7 (other parts) and C25.8 (overlapping). In the Girona Cancer registry, we collect retrospectively all NETs regardless of their behaviour (malign or uncertain) from 1994 until now. All these cases were recoded following the 2010 international recommendation for codifying NETs in cancer registries.

Survival analysis was performed on a dataset excluding cases identified by death certificate only (DCO) and those cases diagnosed at autopsy (*n* = 119 cases). Follow up was defined from data of diagnosis to date of last known vital status (death by any cause, date of loss of follow-up, or date of end of follow-up at 31st December 2018). Vital status of patients was obtained by linking records to the Catalan registry of Mortality and the National Death Index. We estimated 1- and 5-year observed survival (OS) by sex, age-group, histological type (survival was analyzed jointly for both non-PNETs and non-HC PC on the basis that they were more likely to be carcinomas), tumour location and period of diagnosis. We did not calculate survival estimates when subgroups had <20 patients available for analysis.

### 2.3. Study Outcomes and Statistical Analysis

For data analysis, version 3.4.3 of RStudio was used. Statistical significance was defined as two-sided *p* < 0.05. The study focused in two main outcomes: incidence and survival. Incidence was analyzed in terms of crude rate (CR) and estimates were age-standardized with the following weights: world (ASRw), United States 2000 (ASRus2000), European 1976 (ASRE76) and European 2013 (ASRE13) standard populations. For the sake of simplicity, we will mainly refer to ASRE13 in the text. Trends were assessed using the estimated annual percentage of change (APC) of the ASRE13. The jointpoint loglinear regression version 4.5.0.1 model was used to calculate APC. To analyze differences between sexes according to clinical and pathological characteristics, we used the T test or Chi square test. Version 22.0.0 of SPSS and WebSurvCa [14] have been used for survival analysis. Observed survival was calculated using the Kaplan–Meier method. To eliminate the possibility of death from other causes, the relative survival was calculated using the Pohar–Perme estimator [15], which represents the hypothetical survival that patients would have had if their cancer had been the only possible cause of death. To calculate the net survival, we used life tables and general mortality with the Edlant–Johnson method [16].

### 2.4. Ethics Approval

This study involves a secondary data analysis from existing data and records. The information was recorded by the investigators in such a manner that subjects could not be identified, directly or through identifiers linked to the subjects. The participating cancer registry has data management policies in place allowing for the preservation of individual patients’ confidentiality including the ethical approvals from local mandatory bodies. For this type of study, formal consent is not required.

## 3. Results

### 3.1. Baseline Characteristics

Table 1 shows the characteristics of the 1602 patients with PC identified in Girona during 1994–2015. Of them, 46% were women and 54% men. The mean age at diagnosis was 70.57 years (range 19–101), men were younger (68 years) than women (73 years), *p* < 0.001. DCO-cases were 7.5% in men and 7.3% in women. Patients over 64 years old represented 70% of cases (64.51% in males and 76.11% in females). Overall, 48.5% had a pathologically verified PC. Among them, non-PNETs accounted for 91.63% of cases and PNETs 8.37% (4.06% of the whole series). Non-HC PC comprised the largest group of PC in our study (51.5%, ASR_E13_ = 6.8 per 100,000 p-y). Within non-PNETs, the most frequent histology was Adenocarcinoma (Appendix A). Staging information was only available in 613 patients, 415(67.69%) were stage IV.

### 3.2. Incidence and Incidence Temporal Trends

The overall CR for the whole series was 11.43 cases per 100,000 inhabitants/year. Compared with females, males had higher incidence rates: 12.17 versus 10.34, respectively. The overall ASR_E13_ was 13.19 (95% confidence interval (CI) 12.55; 13.85) and the ASR_W_ was 5.68(95% CI 5.38; 6.01) (Table 2). Age-specific rates reflected an increase with advancing age, having the Girona population over 85 years the highest rate (74.5 cases per 100,000 inhabitants/year) (Figure 1). During the study period, the ASR_E13_ gradually increased from 10.6 in 1994 to 14.21 in 2015, with a significant EAPC of 1.55%. This increase was observed in both males (EAPC = 1.35%) and females (EAPC = 1.67%). No differences in incidence rates between the geographical areas of Girona province have been found, and no significant cut-point was identified by joinpoint regression. (Appendix A; Figure 2a–c).

Within the non-PNETs, the incidence trend evidences a gradual increase. Regarding the ASR, results show an ASR_E13_ of 5.87 (95% CI 5.44; 6.32), that heightens with age (Table 2). During the study period, the ASR_E13_ increased with a significant EAPC of 6%. No significant cut-point was identified by joinpoint regression in this subgroup (Figure 2d).

In our study, PNETs represent only 4.06% of the whole series. The CR was 0.5 (95% CI 0.38; 0.63). As for the ASR, results show an ASR_E13_ of 0.53 (95% CI 0.41; 0.68) and an ASR_W_ of 0.33 (95% CI 0.25; 0.45). Rates remain stable during 1994–2015 with a non-statistically significant EAPC of 3.16% (Figure 2e).

The subgroup of non-HC PC represents 51.5% of all cases. A proportion that was higher in females than in males (57.12% vs. 46.71%). Incidence analyses reveal a total ASR_E13_ of 6.83 (95% CI 6.37; 7.32) that hikes up to 45.9 (95% CI 41.9; 50.1) in >75-year-old patients (Table 2). Rates of non-histology confirmed PC decreased through the whole study period (1994–2015) with a significant EAPC of −2.23% (Figure 2f).

The increase in incidence rates applied for all age subgroups in our population, however the rate of increase was more pronounced in older age groups, as reflected by an EAPC of 1.45% in patients ≥55 years old versus 0.59% in <55 year olds (Figure 2g,i). The difference in annual percent change by age group was larger in Non-PNETs, with an EAPC of 0.6% for <55-year-old patients compared to 6.80% in ≥55 year olds (Figure 2h,j).

### 3.3. Survival

The overall 5-year RS was 7.05% (95% CI: 5.63; 8.84), higher in men (7.34%) than in women (6.75%) (Appendix A). The survival rate decreased with advancing age at diagnosis, the 5-year RS dropped from 34.23% in young patients (15–44 years) to 2.04% in older patients (75+ years) (Figure 3 and Figure 4). According to the stage at diagnosis, stage IIA showed a 5-year RS of 46.14% (95% CI 36.14; 58.91) in contrast to the 4.36% (95% CI 2.61; 7.29) of stage IV. Trends in 5-year RS increased from 3.27% (95% CI 1.69; 6.35) in 1994–98 to 13.1% (95% CI 9.98; 17.2) in 2010–15. No differences in survival rates between the geographical areas of Girona province have been found (Table 3 and Figure 3a).

For non-PNETs and non-HC PC, overall 5-year RS was higher in men (4.75%) than in women (2.9%). The survival rate was lower with increasing age, the 5-year RS decreased from 15.79% in young patients (15–44 years) to 1.07% in older patients (75+ years). As for the stage at diagnosis, stage IIA presented the best survival with a 5-year RS of 28.25% (95% CI 14.08; 56.67) and stage IV patients carried the worst 5-year RS: 0.55% (95% CI 0.08; 3.89). Trends in 5-year RS rose from 1.06% (95% CI 0.36; 3.1) in 1994–98 to 7.76% (95% CI 5.22; 11.53) in 2010–15. No differences in survival between the geographical areas studied have been found (Table 4 and Figure 3b).

Finally, patients with PNETs had the best survival with a 5-year RS of 61.45% (95% CI 47.47; 79.55) (Figure 4). Due the low number of cases we only analysed survival according to sex, age group and period of diagnosis. No differences were observed by age or sex. The 5-year RS increased during the study period from 45.2% (95% CI 23.28; 87.77) in 1994–1998 to 70.31 (95% CI 54.71; 90.36) in 2010–2015 (Table 5 and Figure 3c).

## 4. Discussion

We present the first registry-based data on trends in PC incidence and survival rates in Spain for all histological types, including neuroendocrine tumors.

Our results for the Girona province revealed an upward trend in PC incidence throughout the study period regardless of sex, age, tumour location, stage, period of diagnosis and geographical area. In contrast with Wood et al. [7] who reported a stable or even decreasing trend, our findings concur with recent publications that report a stepwise increase of PC incidence in the last decades, especially evident in developed countries [4,6,17,18,19,20]. According to the prevailing worldwide tendency [4], in the current study PC incidence is higher in men than in females and increases with age for both sexes, with 70% of patients being older than 64 years at diagnosis. The aging and longevity of the population, the increasing incidence of obesity and adult-onset diabetes, and exposure to risk factors such as smoking, and some dietary habits (consumption of red and processed meat or excessive alcohol intake, for example) may partially account for the present findings [21].

In our series we found 51.5% of PC without microscopic verification, a higher result than the proportion published in other studies [17]. Obtaining tissue for diagnosis can be notoriously difficult in patients with suspected PC and it often requires invasive methods, such as ultrasound-guided biopsies or even surgery. These procedures can be challenging in elderly patients or in those with poor performance status, reason whereby we believe there is a high proportion of PC with unspecified histology (malignant neoplasm) in our series. This weakness in registry-based studies of PC has been largely reported. In EUROCARE-4 [22], a study that collected data from 93 European cancer registries between 1995 and 2002, the microscopic verification rate was 63% (range 30–91%) for PC. What stands out from our data is the ASR behaviour of the non-HC PC, whose incidence rate gradually decreases with an EAPC of −2.23. These findings not only reflect the introduction and growing use of modern diagnostic techniques in our daily clinical practice, but also a change of attitude towards pancreatic cancer leaving nihilism behind.

There is growing evidence supporting an association between adult weight gain, childhood or adolescent obesity, and increased risk of colorectal, endometrial, and pancreatic cancers [23]. Sung et al. have recently reported an increasing of PC incidence in young adults compared to >45-year-old population, and linked it to excess bodyweight, obesity-related health conditions and lifestyle factors [24]. Unlike them, although in our study PC incidence is growing in all age groups, we found a greater rise of incidence trend in older people (≥55 years old) compared with younger adults (<55 years old), and the difference was more evident when we limited our analysis to non-PNET histologically confirmed cases. This could be explained by two main reasons: first, the low number of cases diagnosed <45 years old (*n* = 52); and, second, although in Spain obesity rates in adolescents and young adults are rising due to changes in life-style and diet [25], it is a recent issue and it is not reflected yet in the PC incidence rates during the study period.

In accordance with previously reported data [5,11,26], most cases (*n* = 613) were diagnosed in stage IV (67.69%). This data reflects the absence of specific symptoms in the early stages of pancreatic tumors. Even in the highest income countries most PC patients are diagnosed with metastatic or locally advanced disease [27], a fact which determines the poor survival of PC patients worldwide.

Survival for PC, as reported in the EUROCARE-5 study [10], is particularly low: European average RS is 26% at one year and 7% at five years since diagnosis, similar to what we found in our series (Figure 3). In accordance with previously published data [8], survival decreased with advancing age in our study, except for PNETs, where we found no differences among age groups.

We observed better survival in males than in females, and this is a previously unreported finding. In Europe, 5-year RS was similar in both sexes when considering European regions or countries [10]. Looking back to the demographic characteristics of our population (Table 1), two important data stand out: the median age of the female cohort is higher than the males’ (75 y vs. 70 y) and the main difference lies in the proportion of population over 75 y at time of PC diagnosis (34% of males and 53% of females). As age is one of the strongest prognostic factors for PC, this may explain the poor outcomes of females in our study.

As expected, due to the different etiology and biology of the PNET neoplasms [28,29], survival was much higher than for the rest of the study population (the non-PNETs). The 5-year RS for this subgroup was 61.45 (95% CI 47.47; 79.55), above those observed in Europe for the period 2000–2007: 42.9% (95% CI 41.0; 44.8) [30]. Following the whole series tendency, it improved thorough the study timeframe. When we approach PNETs’ survival results, it is to note that since 2010, the international recommendation for the registration of PNET in cancer registries has changed. The new stage classification of NETs is based on histological grades, thus since its implementation all NETs should be registered, whereas until then only invasive cases were registered. In Girona all cases were recoded following these recommendations, and due to this fact, PNETs survival may have improved. Unlike in the rest of PC, females presented better survival than males, and survival was similar among the age groups studied.

With regards to survival rates in non-PNETs, irrespective of being microscopically confirmed or not, encouraging progress in survival over time was detected. We did not only find an increase in 1-year RS, but also, in contrast with the published data from the UK [31] and the Canadian Cancer Registry [32], an upswing in the 5-year RS from 1.06% (95% CI 0.36; 3.1) in the initial lustrum (1994–1998) to 7.76% (95% CI 5.22; 11.53) in the last time period of the study (2010–2015). We have an ongoing higher resolution study in order to analyze the treatment regimens received by this population and their impact on survival.

### Strengths and Weaknesses

Strengths of this study include the use of real-world data from a high-quality population-based Cancer Registry, the large sample size, and the long-term and virtually complete follow-up of the population that yield robust survival. However, the study presents some limitations that should be kept in mind when interpreting our results: first the territory-linked data as a result of being the Regional Cancer Registry as our source of information. Second, we had missing information on clinical variables such as AJCC (American Joint Committee on Cancer) stage and tumour location for more than half of the study population. And third, we lacked information on tumour histology in 51% of patients.

### 5. Conclusions

We present a large, comprehensive, and up-to-date analysis of incidence and survival data of all PC in Girona (Spain) obtained from a high-quality cancer registration system. To the authors’ knowledge, this is the first population-based study in Spain to analyse both incidence and survival trends for PC. Our results revealed an upward trend in PC incidence throughout the study period regardless of sex, age, tumour location, stage, period of diagnosis and geographical area, as well as an encouraging increase in survival in recent years. PNETs diagnosis had the best survival rates. Among non-PNETs or non-HC PC, advanced age and stage were associated with poorer survival rates. This information will contribute to estimate the burden of PC in Spain and will serve as a baseline for future analysis, allowing comparison with other registries’ data.

## Figures and Tables

**Figure 1 ijerph-17-09538-f001:**
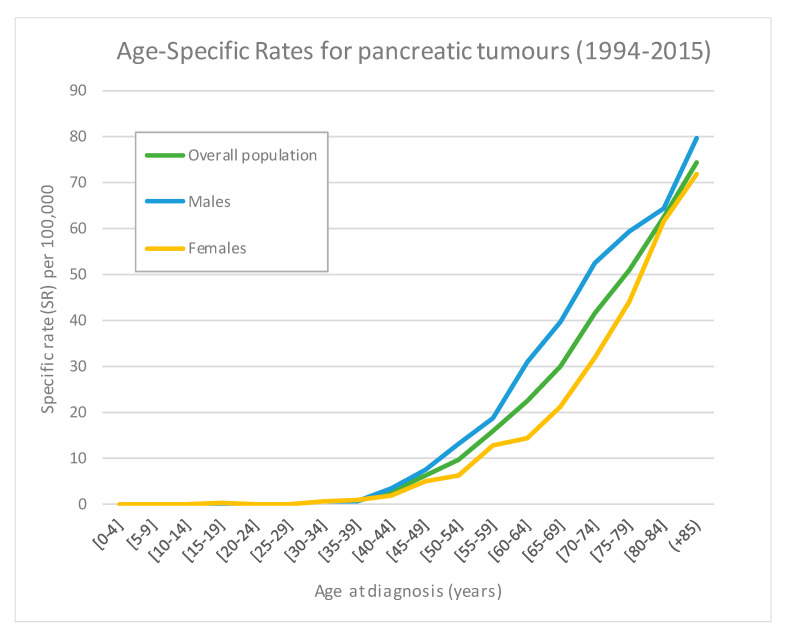
Age-specific incidence rates for pancreatic tumours of the overall population and according to sex.

**Figure 2 ijerph-17-09538-f002:**
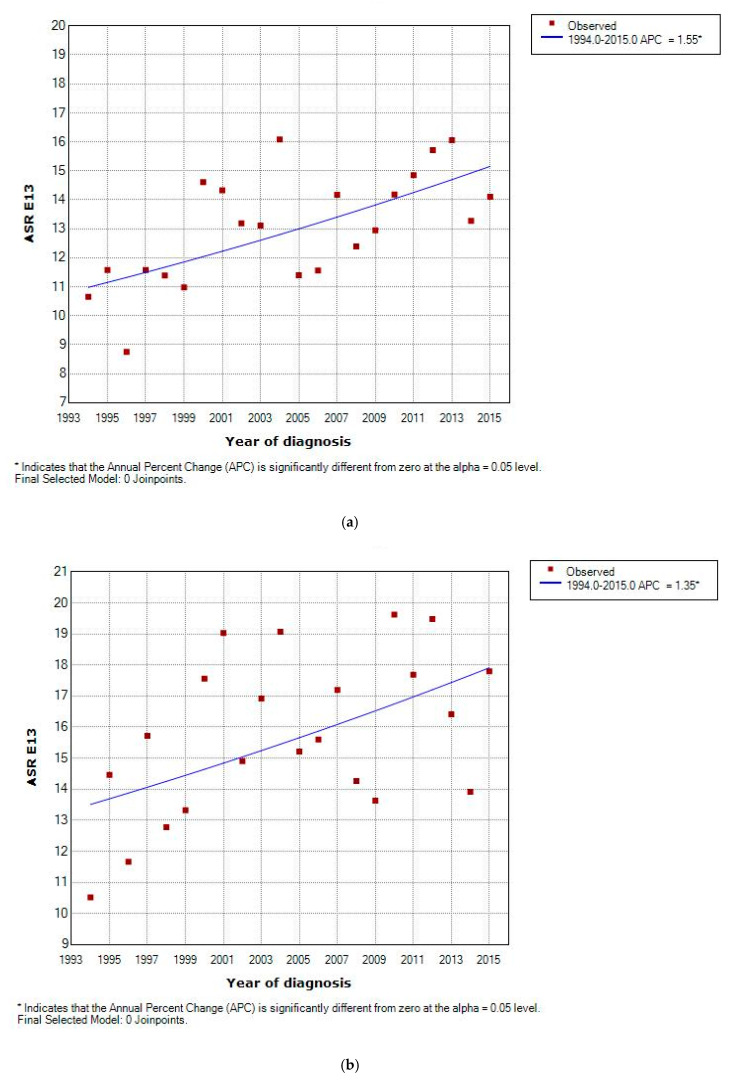
Trends in annual pancreatic tumors incidence (ASR_E13_) according to sex; (**a**) both sexes; (**b**) male; (**c**) female; according to histological groups; (**d**) non PNETS; (**e**) PNETs; (**f**) non histological confirmation. According to age groups; (**g**) All PC ≥ 55 years-old; (**h**) Exocrine tumours ≥ 55 years-old; (**i**) All PC < 55 years-old; (**j**) Exocrine tumours < 55 years-old.

**Figure 3 ijerph-17-09538-f003:**
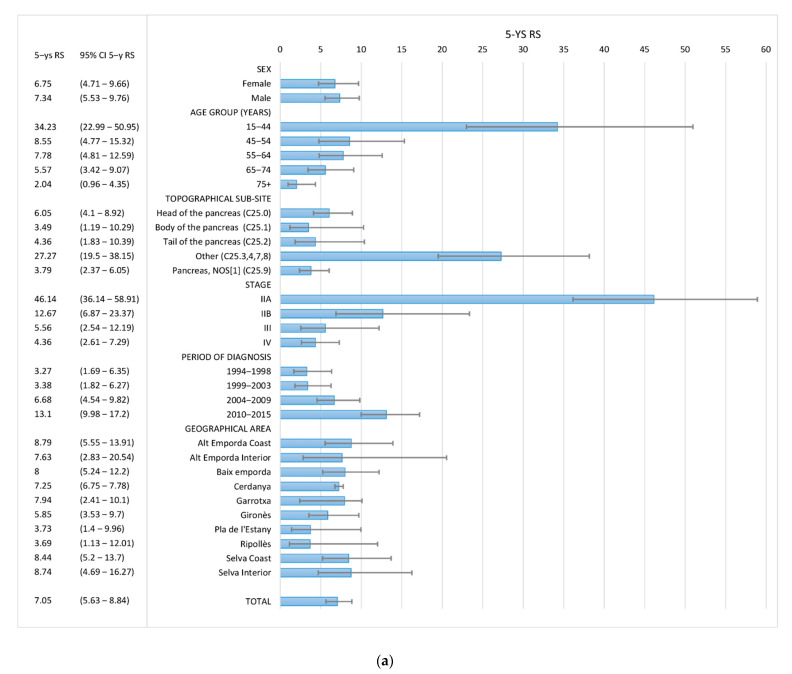
5-year relative survival for all pancreatic cancers—PC (**a**), Pancreatic carcinomas non-neuroendocrine—Non PNETs (**b**) and Pancreatic neuroendocrine tumors—PNETs (**c**) by sex, stage, period of diagnosis, sub-site and county. Girona province, 1994–2015.

**Figure 4 ijerph-17-09538-f004:**
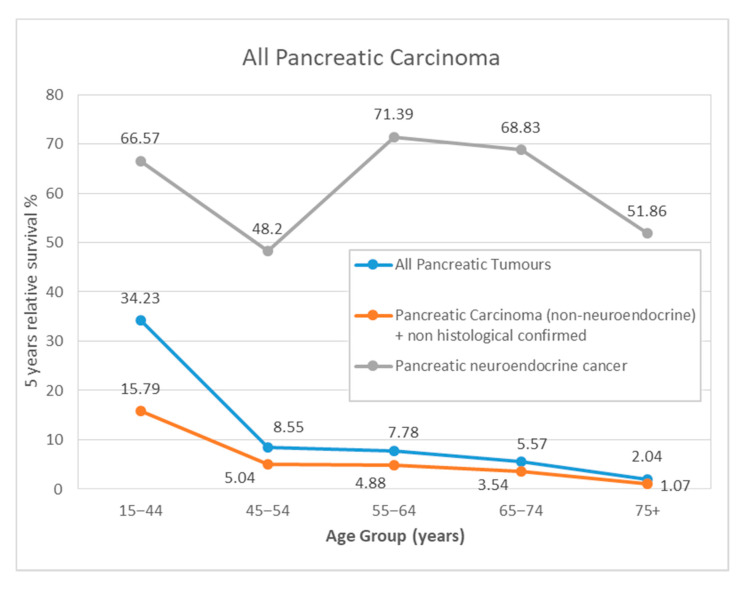
Age-specific period estimates of 5-year relative survival for all patients with pancreatic cancer diagnosed between 1994–2015, and according to histology.

**Table 1 ijerph-17-09538-t001:** Distribution of pancreatic tumours characteristics according to sex. Girona 1994–2015.

	Total	Males	Females	*p*-Value
	*n* (%) (% *)	*n* (%) (% *)	*n* (%) (% *)	
AGE AT DIAGNOSIS ^				<0.001
Mean	70.57	68.49	73.01	
Median	72	70	75	
Min-max	19–101	28–96	19–101	
AGE GROUP (YEARS) ^+^				<0.001
15–44	52 (3.25)	29 (3.35)	23 (3.12)	
45–54	150 (9.36)	98 (11.33)	52 (7.06)	
55–64	281 (17.54)	180 (20.81)	101 (13.7)	
65–74	432 (26.97)	262 (30.29)	170 (23.07)	
75+	687 (42.88)	296 (34.22)	391 (53.05)	
BASIS OF DIAGNOSIS ^+^				<0.001
Death certificate only (DCO)	119 (7.43)	65 (7.51)	54 (7.33)	
Clinical only	4 (0.25)	1 (0.12)	3 (0.41)	
Clinical investigation	642 (40.07)	304 (35.14)	338 (45.86)	
Cytology	251 (15.67)	148 (17.11)	103 (13.98)	
Histology of a metastasis	155 (9.68)	91 (10.52)	64 (8.68)	
Histology of a primary tumour	412 (25.72)	246 (28.44)	166 (22.52)	
Unknown	19 (1.19)	10 (1.16)	9 (1.22)	
TOPOGRAPHICAL SUB-SITE ^+^				0.604
Head of the pancreas (C25.0)	658 (41.07)	357 (41.27)	301 (40.84)	
Body of the pancreas (C25.1)	109 (6.8)	56 (6.47)	53 (7.19)	
Tail of the pancreas (C25.2)	89 (5.55)	51 (5.89)	38 (5.15)	
Other specified parts of the pancreas (C25.3,4,7,8)	61 (3.8)	28 (3.23)	33 (7.47)	
Pancreas, NOS (C25.9)	685 (42.75)	373 (43.12)	312 (42.33)	
HISTOLOGICAL GROUPS ^+^				<0.001
PNETs	65 (4.06)	37 (4.28)	28 (3.8)	
Pancreatic carcinomas (non-neuroendocrine)	712 (44.44)	424 (49.02)	288 (39.08)	
Pancreatic tumours non-histologically confirmed	825 (51.5)	404 (46.71)	421 (57.12)	
TNM STAGE ^+^				0.556
I	21 (1.31) (3.42)	10 (1.15) (3.04)	11 (1.49) (3.84)	
IIA	37 (2.31) (6.03)	21 (2.43) (6.40)	16 (2.17) (5.61)	
IIB	70 (4.37) (11.41)	41 (4.74) (12.5)	29 (3.93) (10.17)	
III	70 (4.37) (11.41)	43 (4.97) (13.1)	27 (3.66) (9.47)	
IV	415 (25.90) (67.69)	213 (24.62) (64.93)	202 (27.41) (70.87)	
Unknown	989 (61.73)	537 (62.08)	452 (61.32)	
PERIOD OF DIAGNOSIS ^+^				0.993
1994–1998	239 (14.91)	128 (14.79)	111 (15.06)	
1999–2003	327 (20.41)	177 (20.46)	150 (20.35)	
2004–2009	453 (28.27)	246 (28.43)	207 (28.08)	
2010–2015	583 (36.39)	314 (36.30)	269 (36.49)	
TOTAL	1602 (100.0)	865(54.2)	737 (45.8)	

PNETs—Pancreatic neuroendocrine tumours. TNM—Tumor node metastasis. (%) * Percentage with respect to the *n* cases with confirmed stage (*n* = 613 for both sexes, *n* = 328 for males, and *n* = 286 for females).^ Results from *t*-test. ^+^ Results from Chi-Square test.

**Table 2 ijerph-17-09538-t002:** Incidence of pancreatic cancer (PC) in Girona, 1994–2015. European standard population of 2013 (ASRE13) according to histology. Non-PNET—Pancreatic carcinomas non-neuroendocrine. PNET—Pancreatic neuroendocrine tumours. Non-HC—Pancreatic cancers non-histologically confirmed.

	All PC				Non-PNETs				PNETs				PC Non-HC			
	*n*	% Total	ASR_E13_	95% CI	*n*	% Total	ASR_E13_	95% CI	*n*	% Total	ASR_E13_	95% CI	*n*	% Total	ASR_E13_	95% CI
Sex																
Female	737	46	10.82	(10.05–11.65)	288	40.45	4.46	(3.95–5.01)	28	43.08	0.41	(0.27–0.6)	421	51.03	5.96	(5.4–6.57)
Male	865	54	15.8	(14.75–16.91)	424	59.55	7.35	(6.66–8.1)	37	56.92	0.6	(0.42–0.84)	404	48.97	7.85	(7.09–8.68)
Age Group (years)																
15–44	52	3.25	0.85	(0.63–1.12)	27	3.79	0.44	(0.29–0.65)	19	29.23	0.3	(0.18–0.48)	6	0.73	0.1	(0.04–0.22)
45–54	150	9.36	8.16	(6.9–9.58)	106	14.89	5.75	(4.7–6.96)	13	20	0.7	(0.37–1.2)	31	3.76	1.71	(1.16–2.43)
55–64	281	17.54	19.32	(17.13–21.72)	191	26.83	13.14	(11.34 15.14)	10	15.38	0.69	(0.33–1.26)	80	9.7	5.5	(4.36–6.85)
65–74	432	26.97	35.66	(32.38–39.19)	221	31.04	18,23	(15.91–20.8)	11	16.92	0.9	(0.45–1.62)	200	24.24	16.52	(14.31–18.98)
75+	687	42.88	61.83	(57.28–66.66)	167	23.46	14.87	(12.69 17.32)	12	18.46	1.06	(0.55–1.86)	508	61.58	45.91	(41.99–50.1)
STAGE																
I	21	1.31	0.15	(0.07–0.32)	9	1.26	0.08	(0.02–0.21)	8	12.31	0.05	(0.01–0.17)	4	0.48	0.03	(0.01–0.14)
IIA	37	2.31	0.31	(0.22–0.43)	25	3.51	0.21	(0.14–0.32)	5	7.69	0.04	(0.01–0.09)	7	0.85	0.06	(0.02–0.12)
IIB	70	4.37	0.59	(0.46–0.75)	62	8.71	0.53	(0.4–0.68)	1	1.54	0.01	(0–0.04)	7	0.85	0.06	(0.02–0.12)
III	70	4.37	0.58	(0.45–0.74)	43	6.04	0.36	(0.26–0.49)	2	3.08	0.01	(0–0.05)	25	3.03	0.21	(0.13–0.31)
IV	415	25.90	3.40	(3.07–3.79)	251	35.25	2.06	(1.8–2.37)	21	32.31	0.16	(0.1–0.28)	143	17.33	1.18	(0.99–1.42)
TOTAL	1602		13.19	(12.55–13.85)	712		5.87	(5.44–6.32)	65		0.49	(0.38–0.63)	825		6.83	(6.37–7.32)

**Table 3 ijerph-17-09538-t003:** One and 5-year observed and relative survival for all pancreatic cancers by sex, age, stage, period of diagnosis, sub-site and county. Girona province, 1994–2015.

	Observed Survival (OS)					Relative Survival (RS)				
	n Cases	1–y OS	95% CI 1–y OS	5–y OS	95 %CI 5–y OS	n Cases	1–y RS	95% CI 1–y RS	5–ys RS	95% CI 5–y RS
**SEX**										
Female	681	20.46	(17.6–23.79)	4.18	(2.76–6.34)	681	26.46	(23.04–30.38)	6.75	(4.71–9.66)
Male	800	22.47	(19.69–25.65)	5.83	(4.28–7.95)	800	24.12	(21.28–27.34)	7.34	(5.53–9.76)
**AGE GROUP (YEARS)**										
15–44	51	54.65	(42.51–70.25)	33.91	(22.57–50.95)	51	54.73	(42.72–70.11)	34.23	(22.99–50.95)
45–54	146	37.62	(30.37–46.61)	8.39	(4.6–15.31)	146	37.75	(30.52–46.68)	8.55	(4.77–15.32)
55–64	262	30.02	(24.8–36.36)	7.37	(4.51–12.05)	259	30.62	(25.33–37.02)	7.78	(4.81–12.59)
65–74	400	23.98	(20.05–28.67)	4.69	(2.82–7.8)	398	24.64	(20.62–29.43)	5.57	(3.42–9.07)
75+	622	9.92	(7.8–12.63)	1.26	(0.57–2.77)	618	10.72	(8.43–13.64)	2.04	(0.96–4.35)
**TOPOGRAPHICAL SUB–SITE**										
Head of the pancreas (C25.0)	644	26.26	(23–29.99)	4.26	(2.77–6.53)	644	30.7	(27.09–34.79)	6.05	(4.1–8.92)
Body of the pancreas (C25.1)	106	19.56	(13.13–29.15)	2.06	(0.52–8.11)	106	19.16	(13.28–27.64)	3.49	(1.19–10.29)
Tail of the pancreas (C25.2)	82	18.05	(11.28–28.88)	4.51	(1.55–13.1)	82	19.9	(13.61–29.09)	4.36	(1.83–10.39)
Other specified parts of the pancreas (C25.3,4,7,8)	91	45.75	(36.53–57.3)	30.84	(22.23–42.79)	91	41.14	(32.93–51.4)	27.27	(19.5–38.15)
Pancreas, NOS (C25.9)	558	12.91	(10.36–16.1)	2.71	(1.56–4.7)	558	15.41	(12.48–19.02)	3.79	(2.37–6.05)
**STAGE**										
I	21	–	(NA–NA)	–	(NA–NA)	21	–	(NA–NA)	0.26	(NA–NA)
IIA	37	62.49	(48.16–81.1)	36.87	(22.79–59.63)	37	66	(53.63–81.23)	46.14	(36.14–58.91)
IIB	69	60.58	(49.81–73.68)	15.4	(7.58–31.31)	69	56.12	(45.76–68.81)	12.67	(6.87–23.37)
III	70	33.34	(23.88–46.54)	4.62	(1.54–13.89)	70	37.55	(29.73–47.43)	5.56	(2.54–12.19)
IV	414	19.11	(15.57–23.45)	2.78	(1.37–5.63)	414	21.1	(17.43–25.53)	4.36	(2.61–7.29)
**PERIOD OF DIAGNOSIS**										
1994–1998	217	16.69	(12.39–22.49)	2.78	(1.26–6.12)	217	18.09	(13.68–23.91)	3.27	(1.69–6.35)
1999–2003	289	14.18	(10.64–18.89)	2.48	(1.19–5.16)	289	16.39	(12.46–21.55)	3.38	(1.82–6.27)
2004–2009	415	23.04	(19.3–27.5)	4.66	(3–7.22)	415	27.44	(23.51–32.02)	6.68	(4.54–9.82)
2010–2015	560	26.45	(22.89–30.55)	9.65	(7.09–13.13)	560	31.45	(27.6–35.83)	13.1	(9.98–17.2)
**GEOGRAPHICAL AREA**										
Alt Empordà Coast	224	21.99	(17.06–28.35)	5.98	(3.42–10.46)	224	26.25	(21.11–32.66)	8.79	(5.55–13.91)
Alt Empordà Interior	38	14.75	(6.25–34.85)	7.38	(2–27.16)	38	13.69	(7.09–26.42)	7.63	(2.83–20.54)
Baix Empordà	262	23.14	(18.49–28.95)	6.03	(3.5–10.4)	262	24.94	(20.23–30.74)	8	(5.24–12.2)
Cerdanya	24	12.5	(4.34–36.03)	4.17	(0.61–28.38)	24	13.87	(8.82–21.83)	7.25	(6.75–7.78)
Garrotxa	129	19.19	(13.42–27.45)	4.26	(1.82–9.96)	129	21.69	(15.77–29.83)	4.94	(2.41–10.1)
Gironès	343	21.09	(17.11–26)	4.02	(2.21–7.33)	343	25.98	(21.85–30.88)	5.85	(3.53–9.7)
Pla De L’Estany	62	26.28	(17.26–40.02)	3.5	(0.9–13.61)	62	34.21	(25.25–46.36)	3.73	(1.4–9.96)
Ripollès	87	17.43	(10.87–27.96)	3.08	(0.81–11.71)	86	18.29	(11.55–28.97)	3.69	(1.13–12.01)
Selva Coast	204	27.88	(22.19–35.01)	5.93	(3–11.73)	204	30.55	(24.71–37.78)	8.44	(5.2–13.7)
Selva Interior	108	14.18	(8.88–22.64)	5.88	(2.61–13.26)	108	17.38	(11.51–26.24)	8.74	(4.69–16.27)
TOTAL	1481	21.54	(19.5–23.79)	5.07	(3.95–6.51)	1481	24.94	(22.72–27.38)	7.05	(5.63–8.84)

**Table 4 ijerph-17-09538-t004:** One and 5-year observed and relative survival for non-PNETs and non-HC PC by sex, age, stage, period of diagnosis, sub-site and county. Girona province, 1994–2015.

	Observed Survival (OS)					Relative Survival (RS)				
	n Cases	1–y OS	95% CI 1–y OS	5–y OS	95 %CI 5–y OS	n Cases	1–y RS	95% CI 1–y RS	5–ys RS	95% CI 5–y RS
**SEX**										
Female	653	17.79	(15.04–21.04)	1.64	(0.82–3.29)	653	23.24	(19.75–27.33)	2.9	(1.61–5.21)
Male	763	20.21	(17.48–23.37)	3.62	(2.4–5.47)	763	21.78	(18.93–25.07)	4.75	(3.25–6.95)
**AGE GROUP (YEARS)**										
15–44	32	37.5	(23.98–58.65)	15.62	(6.59–37.02)	32	37.56	(24.32–58.02)	15.79	(7.07–35.28)
45–54	133	34.35	(26.97–43.75)	4.94	(2.15–11.36)	133	34.47	(27.13–43.81)	5.04	(2.29–11.09)
55–64	252	27.88	(22.68–34.27)	4.6	(2.39–8.88)	249	28.45	(23.19–34.9)	4.88	(2.59–9.19)
65–74	389	22.65	(18.76–27.33)	2.89	(1.46–5.74)	387	23.27	(19.3–28.06)	3.54	(1.87–6.7)
75+	610	8.99	(6.96–11.62)	0.57	(0.16–1.99)	606	9.73	(7.53–12.57)	1.07	(0.37–3.11)
**TOPOGRAPHICAL SUB–SITE**										
Head of the pancreas (C25.0)	638	26.19	(22.92–29.93)	3.94	(2.51–6.16)	638	30.74	(27.11–34.86)	5.73	(3.83–8.58)
Body of the pancreas (C25.1)	105	19.75	(13.26–29.42)	2.08	(0.53–8.18)	105	19.16	(13.28–27.64)	3.5	(1.19–10.29)
Tail of the pancreas (C25.2)	78	13.67	(7.72–24.19)	0	(NA–NA)	78	15.9	(10.03–25.23)	0.05	(0.02–0.17)
Other specified parts of the pancreas (C25.3,4,7,8)	57	18.01	(10.24–31.7)	2.33	(0.35–15.74)	57	20.73	(13.55–31.73)	2.62	(0.7–9.81)
Pancreas, NOS (C25.9)	538	11.44	(9–14.55)	1.83	(0.94–3.59)	538	13.89	(11–17.53)	2.65	(1.46–4.81)
**STAGE**										
I	13	–	(NA–NA)	–	(NA–NA)	13	–	(NA–NA)	–	(NA–NA)
IIA	32	55.98	(40.45–77.47)	25.02	(12.29–50.91)	32	58.06	(42.18–79.92)	28.25	(14.08–56.67)
IIB	68	61.47	(50.65–74.61)	15.63	(7.69–31.75)	68	56.12	(45.77–68.81)	12.67	(6.87–23.37)
III	68	31.35	(22–44.68)	3.2	(0.82–12.44)	68	34.68	(27.1–44.37)	2.89	(0.93–8.93)
IV	393	17.27	(13.81–21.61)	1.86	(0.8–4.37)	393	19.61	(15.92–24.14)	3.48	(1.89–6.4)
**PERIOD OF DIAGNOSIS**										
1994–1998	208	14.52	(10.43–20.21)	0.97	(0.24–3.84)	208	16.05	(11.73–21.97)	1.06	(0.36–3.1)
1999–2003	279	12.5	(9.13–17.11)	1.47	(0.56–3.89)	279	13.95	(10.3–18.9)	2.35	(1.08–5.12)
2004–2009	402	21.52	(17.83–25.97)	2.78	(1.56–4.99)	402	25.93	(21.87–30.75)	4.23	(2.51–7.14)
2010–2015	527	22.8	(19.33–26.89)	5.42	(3.44–8.55)	527	27.53	(23.63–32.08)	7.76	(5.22–11.53)
**GEOGRAPHICAL AREA**										
Alt Empordà Coast	216	20.5	(15.65–26.87)	4.53	(2.34–8.79)	216	25.37	(19.95–32.26)	7.43	(4.24–13.03)
Alt Empordà Interior	38	14.75	(6.25–34.85)	7.38	(2–27.16)	38	13.69	(7.09–26.42)	7.63	(2.83–20.54)
Baix Empordà	242	18.38	(14.02–24.08)	1.25	(0.34–4.66)	242	19.24	(14.87–24.88)	2.1	(0.81–5.46)
Cerdanya	23	8.7	(2.31–32.69)	0	(NA–NA)	21	9.83	(3.09–31.3)	0.22	(0.02–2.58)
Garrotxa	124	16.68	(11.2–24.83)	1.81	(0.46–7.07)	124	19.13	(13.43–27.23)	1.88	(0.7–5.01)
Gironès	328	18.41	(14.58–23.25)	1.69	(0.62–4.61)	328	23.54	(19.45–28.48)	2.96	(1.38–6.35)
Pla De L’Estany	58	24.62	(15.63–38.79)	1.89	(0.27–13.14)	58	36.22	(26.85–48.84)	1.75	(0.44–6.92)
Ripollès	83	15.79	(9.43–26.42)	1.48	(0.21–10.22)	82	16.53	(10.03–27.24)	1.85	(0.4–8.55)
Selva Coast	199	26.49	(20.84–33.66)	4.53	(2.1–9.78)	199	28.54	(22.78–35.76)	5.83	(3.17–10.72)
Selva Interior	105	12.66	(7.62–21.01)	4.17	(1.55–11.24)	105	15.54	(9.99–24.17)	6.83	(3.3–14.13)
TOTAL	1416	19.08	(17.09–21.3)	2.71	(1.9–3.87)	1416	22.31	(20.05–24.83)	4.01	(2.88–5.59)

**Table 5 ijerph-17-09538-t005:** One and 5-year observed and relative survival for PNETs by sex, age, stage, period of diagnosis, sub-site and county. Girona province, 1994–2015.

	Observed Survival (OS)					Relative Survival (RS)				
	n Cases	1–y OS	95% CI 1–y OS	5–y OS	95 %CI 5–y OS	n Cases	1–y RS	95% CI 1–y RS	5–ys RS	95% CI 5–y RS
**SEX**										
Female	28	81.82	(68.57–97.62)	68.33	(52.12–89.56)	28	63.63	(55.42–73.06)	61.17	(51.85–72.17)
Male	37	67.27	(53.65–84.34)	51.38	(36.97–71.42)	37	65.32	(51.12–83.45)	53.65	(37.32–77.12)
**AGE GROUP (YEARS)**										
15–44	19	83.88	(68.71–100)	66.09	(47.14–92.65)	19	84	(69.21–101.94)	66.57	(48.06–92.21)
45–54	13	69.23	(48.19–99.47)	47.47	(24.93–90.4)	13	69.39	(49.09–98.07)	48.2	(26.65–87.19)
55–64	10	80	(58.68–100)	68.57	(44.47–100)	10	80.36	(59.95–107.72)	71.39	(47.32–107.72)
65–74	11	71.59	(48.84–100)	61.36	(37.69–99.92)	11	72.56	(50.57–104.1)	68.83	(43.57–108.74)
75+	12	58.33	(36.16–94.1)	43.75	(20.86–91.77)	12	61.42	(39.06–96.58)	51.86	(25.51–105.44)
**TOPOGRAPHICAL SUB–SITE**										
Head of the pancreas (C25.0)	6	33.33	(10.75–100)	33.33	(10.75–100)	6	33.51	(12.7–88.41)	34.11	(12.93–90)
Body of the pancreas (C25.1)	1	0	(NA–NA)	0	(NA–NA)	1	0.03	(0–0.22)	0.03	(0–0.22)
Tail of the pancreas (C25.2)	4	100	(100–100)	75	(42.59–100)	4	100	(100–100)	82.47	(50.94–133.54)
Other specified parts of the pancreas (C25.3,4,7,8)	34	91.18	(82.12–100)	76.55	(62.56–93.67)	34	87.77	(75.23–102.4)	81.77	(64.89–103.02)
Pancreas, NOS (C25.9)	20	52.87	(34.4–81.24)	32.9	(16.24–66.65)	20	60.11	(46.4–77.86)	39.95	(23.52–67.84)
**STAGE**										
I	8	–	(NA–NA)	–	(NA–NA)	8	–	(NA–NA)	–	(NA–NA)
IIA	5	100	(100–100)	100	(100–100)	5	100	(100–100)	106.46	(106.46–106.46)
IIB	1	0	(NA–NA)	0	(NA–NA)	1	0.03	(0–0.22)	0.03	(0–0.22)
III	2	100	(100–100)	50	(12.5–100)	2	100	(100–100)	57.71	(18.81–177.01)
IV	21	51.95	(34.31–78.66)	28.05	(11.96–65.81)	21	51.45	(36.42–72.7)	34.83	(20.62–58.85)
**PERIOD OF DIAGNOSIS**										
1994–1998	9	66.67	(42–100)	44.44	(21.41–92.27)	9	67.18	(43.7–103.27)	45.2	(23.28–87.77)
1999–2003	10	60	(36.17–99.52)	30	(11.64–77.32)	10	60.38	(37.63–96.87)	32.95	(14.24–76.28)
2004–2009	13	69.23	(48.19–99.47)	61.54	(40.04–94.58)	13	69.8	(56.02–86.96)	68.08	(54.17–85.57)
2010–2015	33	81.38	(68.98–96.02)	72.66	(57.68–91.51)	33	81.2	(68.9–95.7)	70.31	(54.71–90.36)
**GEOGRAPHICAL AREA**										
Alt Empordà Coast	8	62.5	(36.54–100)	46.88	(21.49–100)	8	63.07	(38.45–103.46)	48.1	(24.06–96.17)
Alt Empordà Interior	20	79.69	(63.73–99.63)	67.61	(49.04–93.22)	20	73.82	(56.89–95.8)	59.81	(42.09–84.98)
Baix Empordà	1	100	(100–100)	100	(100–100)	1	100	(100–100)	100	(100–100)
Cerdanya	5	80	(51.61–100)	60	(29.33–100)	5	80.32	(54.1–119.23)	62.04	(33.12–116.23)
Garrotxa	15	80	(62.12–100)	54	(32.22–90.51)	15	78.89	(64.45–96.56)	63.59	(48.16–83.97)
Gironès	4	50	(18.77–100)	25	(4.58–100)	4	50.02	(22.1–113.22)	25.07	(7–89.75)
Pla De L’estany	4	50	(18.77–100)	50	(18.77–100)	4	52.94	(23.63–118.62)	58.18	(25.97–130.34)
Ripollès	5	80	(51.61–100)	80	(51.61–100)	5	80.84	(54.53–119.86)	82.27	(55.49–121.98)
Selva Coast	3	66.67	(29.95–100)	66.67	(29.95–100)	3	66.88	(34.83–128.44)	68.59	(35.72–131.71)
Selva Interior	8	62.5	(36.54–100)	46.88	(21.49–100)	8	63.07	(38.45–103.46)	48.1	(24.06–96.17)
TOTAL	65	73.6	(63.57–85.21)	58.69	(47.28–72.84)	65	70.28	(58.67–84.19)	61.45	(47.47–79.55)

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
