# Peer review of "Incidence and Survival Trends of Pancreatic Cancer in Girona: Impact of the Change in Patient Care in the Last 25 Years"

_ijerph, 2020, doi:10.3390/ijerph17249538_

Round 1

Reviewer 1 Report

This study aimed at describing the incidence and survival trends for PC over the last 25 years in a Spanish Girona administrative area.

The topic of pancreatic cancer is particularly of interest due to its worldwide increase in incidence.

This paper is clearly written and relays on a long-standing population-based cancer registry, that is gold standard for descriptive epidemiology. Statistical methods are adapted.

Yet, I’ve some remarks:

Information on stage at diagnosis is lacking. Authors write “Staging information was available for 613/1602 patients”. Pancreatic cancer are not often surgically resected, so that the pTNM stage (relying on pathological report) does not exist in at least two thirds of cases. Yet, the metastatic status at diagnosis can be available in cancer registries (and thus completed for “almost” all cases to be used).

Authors must explain how cancer extension was collected and registered. Under present conditions, results using pTNM stage should not be presented (62% of missing data) (table 1 and figures). May be only M+/M- should be presented?

Since 2010, international recommendation for the registration of NET in cancer registries changed. The new stage classification of NETs is based on histological grades. This new rule strongly modified the registration in database. Since that date (since its implementation in each registry) all NETs should be registered (Grade 1, 2, 3…) whereas until then were only registered invasive cases.

As a consequence, incidence and survival are supposed to mechanically increase after the implementation of this rule in database (recent inclusion of cases previously considered “non invasive and benign” and thus not registered).

This point should be discussed in the methods section (which classification was used over time?) and accordingly, in the discussion.

The proportions of patients by period and sex (table 1) are not informative as depending on demographic characteristics of population (standardized incidence rates over time are informative)

In table 3a and b, I wonder if the results according to geographic area are necessary.

Due to small number of cases, Table 3 c and fig 3 c must be re-considered and strongly limited to some rates in the text after grouping the patients.

I regret the choice of European standard population rates in tables because I’m one of those who think that for international publications (and thus comparisons) only one standard should be used (the world standard is the most used worldwide)

In the end, this study remains quite interesting as it gives epidemiological reference value that are necessary.

Reviewer 2 Report

Pancreatic cancer, previous an uncommon occurrence, is currently a major health problem in the EU and USA because its rapid increased incidence in the past generation.  Understanding the characteristics, and lifestyle attributes, and specific intrinsic risk factors, such as genetics, or extrinsic factors, such as regional or environmental chemical exposures is important to identify risk factors underlying susceptibility or, importantly, resistance, to developing PC in European populations. This study is the initial step to start the fundamental biomedical and epidemiological research to stem the tide of this rapidly growing disease before it becomes endemic. This is excellent work I want your paper to be robustly cited. There are few things listed in recommended changes to ensure its acceptance and use in the biomedical community. Thank you for such a well written paper and timely paper about a very deadly disease that is poised to become an epidemic unless public health measures are taken by the population.

Experimental design is robust with enough detail to replicate and specific descriptions of data that were excluded and the reason for curtailing certain types of data. Statistics is mostly appropiate, I appreciate the appropriate usage of loglinear models albeit Chi-square is best for determining the differences between sexes istead of T-test.

Recommended Corrections:

  • Minor Grammar and Word Choices

Line 51 “gloomy” is a word more appropriate for poor weather, such as a “gloomy English summer” but inadequate for projected health outcomes. For example substitute with the words: grave, ominous, grim, bleak

  • Required change: indicate on the figures, figure captions, and tables when specifically the chi-square vs. T-test was used to compare sexes.

  • Table 1 – correct the word p Value (the “e” falls on the second page and difficult to readers to understand and portrays an unprofessional quality)

  • Figure 1 - Enlarge the legend inside the graph because it is too small to read clearly

  • Figure 2 – x-axis title needs to be larger because it is too small to read clearly and the footnote is not legible (make large)

  • Figure 3 – The numbers must be at least 10 font size to be legible. You have an excellent paper, make your data notable and acceptable to the scientific community by making it large enough to read and evaluate. Many readers will simply skip or throw away your results, analyses and conclusion simply because of the difficult graphs. You and your team clearly put much time and effort into this work, don’t undermine your hard effort and work with poor, illegible figures.

  • Figure 4 – Great example – emulate this graph for the other figures. This figure had the minimum size and correct color combinations to read the data clearly. Remember a much greater number than normal (~10-15%) of male scientists and physicians are color-blind and you need to choose colors in your graphs these individual can distinguish.
